# Thermodynamic Properties and DFT Study on Highly Frustrated Cr_3_BO_6_: Coexistence of Spin-Singlets with Long-Range Magnetic Order

**DOI:** 10.3390/ma16247662

**Published:** 2023-12-15

**Authors:** Ekaterina S. Kozlyakova, Vladimir V. Korolev, Peter S. Berdonosov, Sergey I. Latushka, Nadezhda A. Lyubochko, Alexander N. Vasiliev

**Affiliations:** 1Functional Quantum Materials Laboratory, National University of Science and Technology MISIS, 119049 Moscow, Russia; kozliakova.es@misis.ru (E.S.K.); berdonosov@inorg.chem.msu.ru (P.S.B.); 2Lomonosov Moscow State University, 119991 Moscow, Russia; 3SSPA “Scientific-Practical Materials Research Centre of NAS of Belarus”, 220072 Minsk, Belarus; smer444@mail.ru (S.I.L.); nadzeya.lubochko@gmail.com (N.A.L.)

**Keywords:** borates, magnetic properties, density functional theory

## Abstract

The triangle-based magnetic subsystem of borates with the mineral norbergite structure M_3_BO_6_ (M = Fe, Cr, V) makes these compounds unique to investigate rare quantum ground states influenced by strong magnetic frustration. In this work, we investigated the thermal and magnetic properties of Cr_3_BO_6_ to find that despite very large negative Weiss temperature Θ = −160.7 K, it orders only at *T*_N_ = 4.5 K and experiences a spin-flop transition at µ_0_*H* = 5 T. Density functional theory (DFT) calculations of exchange interaction parameters allow for suggesting the model of magnetic subsystem in chromium borate Cr_3_BO_6_. The results prove the decisive role of magnetic frustration on the formation of long-range order, providing therefore a basis for future study. Both experimental data and first-principles calculations point to the coexistence of chromium spin-singlets with long-range antiferromagnetic order.

## 1. Introduction

Frustrated magnetism can be found in the materials with competing exchange interactions between localized magnetic moments so that all exchanges cannot be satisfied simultaneously. This leads to a strong degeneration of the quantum ground state of the magnetic subsystem. Under certain conditions, this degeneracy can lead to a spin liquid ground state—a very peculiar many-particle entangled quantum system in which localized magnetic moments, on the one hand, are connected by strong exchange interactions, and on the other hand, are not magnetically ordered down to zero temperature [1]. The simplest example of a frustrated magnetic system is the antiferromagnetic triangle in the Ising model—in the case of strong axial magnetic anisotropy it is impossible to satisfy all three antiferromagnetic exchange interactions; as a result, the system turns out to be degenerate and frustrated [2]. More complex magnetic subsystems based on triangles also turn out to be frustrated [3,4,5].

Borates of transition metals represent a poorly studied class of magnetic materials, although the tendency of borates to form complex motifs of cations is commonly known in chemistry [6]. The features of crystal structure allow for predicting exotic frustrated magnetic quantum ground states for this class of compounds. Thus, the triangle-based magnetic subsystem of iron borate Fe_3_BO_6_ with the norbergite structure is a rare example of a weak ferromagnet (also called a canted antiferromagnet) with a magnetic ordering temperature T_C_ = 508 K, in which a magnetic spin-reorientation transition at T_SRT_ = 415 K is the first-order phase transition (or, according to other sources, an intermediate-order phase transition) between different spin structures [7,8]. In most magnetic materials, such reorientation phase transitions are of the second order, which makes this compound and its isostructural modifications unique objects for studying phase transitions caused by magnetic frustration.

Among isostructural compounds such as iron borate with the norbergite structure, currently, there have only been synthesized borates of chromium Cr_3_BO_6_ [9] and vanadium V_3_BO_6_ [10]. Isostructural borates of other transition metals are not mentioned in the literature. However, the obtained samples are contaminated with chromium and vanadium oxides, respectively. Their purity is only 70–90%, which makes it difficult to experimentally study the magnetic properties until samples of better quality are obtained. A more detailed consideration of the thermodynamics of the formation of these phases explains the difficulties in the synthesis of pure samples: the phases are metastable under normal conditions. The calculated enthalpy of phase formation of Cr_3_BO_6_ and V_3_BO_6_ exceeds the enthalpy of mixtures CrBO_3_ + Cr_2_O_3_ or V_2_O_3_ + VBO_3_ by 0.183 eV/atom and 0.109 eV/atom, correspondingly [11,12]. The metastability of phases does not necessarily mean that they cannot be synthesized; there are many opposite examples—from diamond to ε-Fe_2_O_3_ [13]. However, among the known metastable phases, the excess of compound enthalpy over the equilibrium system enthalpy is often lower than 100 meV, so rare phases such as ε-Fe_2_O_3_ with a high enthalpy of phase transformation to hematite of about 0.7–0.8 eV/atom can only be obtained in the nanoform or stabilized by impurities, which is not suitable for magnetic studies [14,15]. Thus, obtaining pure phases of chromium Cr_3_BO_6_ and vanadium V_3_BO_6_ borates for experimental research of their magnetism is a complex synthetic problem. In the absence of pure samples, the theoretical prediction of their magnetic subsystem becomes a useful tool for assessing the perspectives of these compounds for condensed matter physics. In this work, we investigated thermal and magnetic properties and carried out a density functional theory (DFT) study of magnetic exchange interactions in order to model the magnetic subsystem of the chromium borate Cr_3_BO_6_ with the norbergite structure. 

### 1.1. Synthesis and Crystal Structure

The preparation of the Cr_3_BO_6_ sample was performed using a technique similar to the one described in Ref. [9]. Precisely 4.1999 g (0.012 mmol) Cr(NO_3_)_3_ × 9H_2_O (Reachem, reagent grade) and 1.5071 g (0.024 mmol) boric acid (Reachem, ultrapure) were mixed and ground in the agate mortar. The mixture was loaded into a silica glass crucible and heated for 72 h at 680° with one intermediate regrinding. The prepared powder sample, as tested by powder XRD (Stoe STADI-P diffractometer, CuK_a1_ radiation, WINX^POW^ software STOE & Cie GmbH, ICDD PDF2 was used as a reference). The formation of the target Cr_3_BO_6_ compound was observed but with a small number of additional reflexes on the X-ray pattern. The sample was washed with hot deionized water in the beaker, filtered, and dried in air. The powder XRD of the final product shows the pure Cr_3_BO_6_ formation. The X-ray pattern was fully indexed in the orthorhombic system, assuming space group *Pnma* (#62) with cell constants *a* = 9.884(10) Å, b = 8.415(6) Å, *c* = 4.425(6) Å, and cell volume = 368.1(9) Å^3^, which is in agreement with previously published results [9].

In the structure of Cr_3_BO_6_, there are two chromium positions in distorted oxygen octahedra, 8*d* (Cr1—blue) and 4*c* (Cr2—purple), as shown in Figure 1a. Through analysis of the shortest distances between magnetic cations in the crystal lattice, it is possible to identify the layers of triangular zigzag chains of [CrO_6_]-octahedra with common edges. In isosceles triangles [Cr_3_O], the central oxygen is common for three [CrO_6_]-octahedra; the distances in such triangles are *d* (Cr1Cr2) = 3122 Å for the edges and *d* (Cr1Cr1) = 3049 Å for the base. Isosceles triangles are connected via Cr1 atoms at the base, as shown in Figure 1b. Chains are connected to form planes by [BO_4_] borate groups through Cr2 vertices. The planes are shifted relative to each other by half of the *a + c* translation and are connected by borate groups.

The magnetic subsystem Cr_3_BO_6_ consists of Cr^3+^ cations, electronic configuration 3*d*^3^, and spin *S* = 3/2. For the discussion of magnetic properties, it is reasonable to take into account only magnetic exchange interactions between the nearest chromium ions. The magnetic subsystem can then be described using six different exchange interaction parameters J_1_–J_6_, as shown in Figure 1b,c, and also in Table 1: J_1_—between the atoms of the bases of the neighboring triangles, J_2_—between the vertex atom and the base atoms of the triangle and J_3_—between atoms at the base of the triangle, and J_4_–J_6_—between the chains. The J_4_ and J_6_ exchanges represent the mean interchain exchanges. The corresponding Cr2–Cr1 distances are slightly unequal. We justify the use of mean exchanges by the fact that the corresponding distances are much longer than intrachain distances, so we expect relatively low values of J_4_–J_6_.

Each Cr1 cation couples with the surrounding magnetic centers by one of the J_1_–J_3_ magnetic exchanges and two of the J_4_–J_6_ exchanges with the neighboring planes. Each Cr2 cation couples with the surrounding magnetic centers by two exchanges J_2_ and four magnetic exchanges J_4_, J_6_ with the neighboring planes.

### 1.2. Magnetization

The temperature dependencies of *dc* magnetic susceptibility, or reduced magnetization, χ(*T*) = *M*/*H*, at a probe field of μ_0_*H* = 0.1 T, and field dependencies of magnetization *M*(*H*) up to 9 T were measured on pressed powder samples of Cr_3_BO_6_ in the temperature range of 2–300 K using the vibrating sample magnetometer option of the “Quantum Design” Physical Properties Measurements System PPMS-9T. The *ac* measurements of magnetic susceptibility were performed on ACMS option of PPMS-9T at a field of magnitude 3 Oe in the frequency range 10 Hz–10 kHz. There are no signs of frequency dependence of magnetization under those conditions.

The χ(*T*) curves were measured in the zero-field-cooled (ZFC), field-cooled-warming (FCW), and field-cooled-cooling (FCC) protocols, as shown in the inset of Figure 2. The fitting of the χ(*T*) curve, measured in the FCW regime by the Curie–Weiss law,
(1)$$\chi =\chi_{0}+\frac{C_{CW}}{T-\theta_{CW}},$$
is shown in the main panel of Figure 2. In the range 250–300 K, it gives the Curie constant *C*_CW_ = 1.93 emu K mol^−1^ and the Curie–Weiss temperature θ_CW_ = −160.7 K. 

According to the expression,
8*C_CW_* = *ng*^2^*S*(*S* + 1), (2)
where spin *S* = 3/2 for Cr^3+^ cations and the *g*-factor *g* = 2; this value of *C*_CW_ corresponds to only one Cr^3+^ ion (*n* = 1) per formula unit. Since there are three (*n* = 3) Cr^3+^ ions per formula unit in Cr_3_BO_6_, we should assume that two of them are magnetically silent below room temperature, being coupled into spin-singlets with strong antiferromagnetic interaction. Magnetic susceptibility sharply drops at Neel temperature T_N_ = 4.5 K in FCC regime (or 4.8 K in FCW and ZFC regimes), where the long-range magnetic order occurs.

The ratio of the Curie–Weiss temperature θ_CW_ to the Neel temperature *T*_N_,
*f* = θ_CW_/*T*_N_(3)
is the measure of the degree of frustration. In Cr_3_BO_6_, the frustration ratio is extremely large, i.e., *f* = 35.7.

From Pascal’s table, the diamagnetic susceptibility χ_D_ was found to be about −10^−4^ emu/mol, but the Curie–Weiss fitting by Equation (1) gives the large positive value of temperature-independent magnetic susceptibility χ_0_ ≈ 2.0 × 10^−3^ emu/mol. It can also be interpreted as if at room temperature the χ(*T*) curve is not paramagnetic one, but already contains the contribution of the strongly coupled spin dimers.

Below 150 K, an upward deviation from the Curie–Weiss law is observed in the χ(*T*) curve, which indicates the presence of ferromagnetic exchange interactions in the system prevailing over the temperature fluctuations. Such ferromagnetic (FM) deviations in the system with dominant antiferromagnetic (AFM) exchange interactions are typical to many geometrically frustrated materials. This differs from what should be expected for a purely antiferromagnetic two-dimensional magnetic system undergoing short-range magnetic ordering [17].

The FCW, FCC and ZFC curves slightly differ from each other. There is a shift of Neel temperature between the FCC and FCW curves, from 4.5 to 4.8 K, respectively, as shown in the inset of Figure 2. The Neel temperatures, *T*_N_, on ZFC and FCW curves are the same, but the signal of ZFC magnetization is slightly lower near and under the phase transition. The difference in behavior of ZFC, FCC and FCW magnetization could be attributed to the effects caused by the occurrence of competitive magnetic interactions. The cooling in magnetic field leads to competition between different magnetic subsystems (FCC curve), but when the magnetic order is established, it requires the same energy to return the system to paramagnetic disorder (FCW and ZFC).

The field dependencies of magnetization, *M*(*H*), were measured at various temperatures below and above the magnetic ordering temperature, as shown in Figure 3. While at high temperatures, the *M*(*H*) curve seems linear, it deviates from linearity below *T*_N_. At 10 K, the derivative *dM*/*dH* evidences no sign of phase transitions caused by magnetic field, but below *T*_N_, the spin-flop phase transition was observed at about µ_0_*H* = 5 T at 2 K, and 4.5 T at 4 K. 

### 1.3. Specific Heat

The specific heat of Cr_3_BO_6_ is shown in Figure 4, which evidences a second-order phase transition that is seen as a λ-type anomaly on the *C*_P_(*T*) curve at 4.5 K. The latter agrees with *T*_N_ = 4.5 K, determined from the magnetization measurements. Above *T*_N_, the specific heat smoothly rises, reaching a value at room temperature significantly lower than the Dulong–Petit limit of 249.5 J mol^−1^ K^−1^. The high-temperature region of the heat capacity can be fitted by the sum of Debye and Einstein functions,
(4)$$C={a_{D}C}_{D}+{a_{E}C}_{E}=9a_{D}R({T/\theta_{D})}^{3}\int_{0}^{\theta_{D}/T}\frac{x^{4}\exp(x)}{{\left[\exp\left(x\right)-1\right]}^{2}}dx+a_{E}R\frac{{\left(\theta_{E}/T\right)}^{2}\exp(\theta_{E}/T)}{{[\exp(\theta_{E}/T)-1]}^{2}},$$
where *R* is a gas constant, θ_D_ is the Debye temperature, and θ_E_ is the Einstein temperature, which results in θ_D_ = 346 ± 3 K, and θ_E_ = 729 ± 4 K, with the corresponding weights *a*_D_ = 3, *a*_E_ = 7. After the subtraction of this lattice heat capacity, the resulting magnetic entropy at *T* < 50 K (12.4 J mol^−1^ K^−1^) is nearly three times lower than the theoretical value for three magnetic Cr^3+^ ions 3*R*ln4 = 34.6 J mol^−1^ K^−1^. A possible explanation is that the absent magnetic entropy was released at higher temperatures due to the formation of AFM spin-singlets (cf. Cr1-Cr1 exchange J_3_, as proposed in the DFT calculations section). Thus, the magnetic entropy released at low temperatures of magnetic ordering transition could be compared with the theoretical value for the one Cr^3+^ ion, *R*ln4 = 11.5 J mol^−1^ K^−1^.

Overall, the thermal *C*_P_(*T*) and magnetic χ(*T*) measurements of Cr_3_BO_6_ revealed a strong deficiency in the effective magnetic moment and magnetic specific heat, which points to the formation of the spin-singlets at elevated temperatures. These singlets coexist with the long-range antiferromagnetic order at about helium temperature. The long-range order concerns only one-third of the chromium ions present in the system.

### 1.4. DFT Calculations

The calculations were carried out by the density functional theory (DFT) method using the Quantum Espresso open-source software package operating on the plane wave basis [18,19,20]. The research was carried out using the supercomputer equipment “Lomonosov 2” of the shared research facilities of HPC computing resources at Lomonosov Moscow State University [21]. The experimental crystal structure data on chromium borate, without additional relaxation, were used in the calculations. For Cr, B, and O atoms, ultrasoft pseudopotentials were chosen from the SSSP PBE Efficiency v1.3.0 and SSSP PBEsol Efficiency v1.3.0 collection [22,23]. The generalized gradient functional (GGA) of Perdue–Burke–Ernzerhof, PBE [24], and PBEsol [25] functionals were used. An automatically generated 4 × 4 × 6 reciprocal lattice grid was used for integration over the Brillouin zone. To consider the Coulomb interactions and correlation effects of the 3*d* electrons in chromium atoms, the on-site Hubbard potential U = 4 eV was applied. To further investigate the effect of the value of the Hubbard potential, it was refined using density functional perturbation theory (DFPT) [26]. The value U = 6.5 eV was obtained on a 2 × 2 × 2 q-grid.

The cut-off kinetic energy for plane waves was chosen to be 680 eV (50 Ry). An analysis of the numerical stability of the solutions was performed to ensure that any further increase in the cut-off energy would not change the qualitative and quantitative picture of the magnetic exchanges. 

To evaluate the amplitude of exchange interactions, a series of single-point calculations of the total energy of various magnetic sublattice configurations were carried out, as shown in Figure 5. Highly asymmetric states (E8, E9) were used to estimate J_3_ coupling. These calculations assume that the exchange integrals are isotropic, and the magnetic subsystem is assumed to be collinear. Equation (5) represents the Hamiltonian for the unit cell spin system:(5)$$\hat{H}=-\sum_{i<j}J_{ij}S_{i}S_{j}=\;={-[J}_{1}\left(\sum_{4}S_{C{r1}_{A}}S_{C{r1}_{B}}\right)+J_{2}\left(\sum_{8}S_{C{r2}_{A}}S_{C{r1}_{B}}\right)+J_{3}\left(\sum_{4}S_{C{r1}_{A}}S_{C{r1}_{B}}\right)+J_{4}\left(\sum_{16}S_{C{r2}_{A}}S_{C{r1}_{B}}\right)\;+J_{5}\left(\sum_{8}S_{C{r1}_{A}}S_{C{r1}_{B}}\right)+J_{6}\left(\sum_{16}S_{C{r2}_{A}}S_{C{r1}_{B}}\right)]$$

Cr1 and Cr2 correspond to 8d and 4c positions, and indexes A and B run among the atoms of unit cell according to Figure 1c for each J. 

The magnetic exchange interaction values were calculated by fitting the overdetermined system of the linear equations, presented in Figure 5, using the least squares method. The results are shown in Table 2. Different columns correspond to different calculations parameters—the functional and Hubbard potential used for chromium 3*d* electrons. The main antiferromagnetic exchange interaction is J_1_, connecting the [Cr_3_O] triangles in the chain. Exchange coupling, J_3_, at the base of the [Cr_3_O] isosceles triangle, is also found to be antiferromagnetic, which leads to frustration in the aforementioned triangle. Frustration persists regardless of the sign of the rather small J_2_ exchange along the edges of the [Cr_3_O]. The resulting interchain magnetic exchange interactions J_4_, J_5_, and J_6_ were small, compared to J_1_, and predominantly ferromagnetic, with J_4_ and J_6_ being frustrated. 

### 1.5. Magnetic Model

Considering the hierarchy of exchange interaction based only on its amplitude, the magnetic structure is based on a system of interacting AFM dimers via J_1_, connected in chains by J_3_ and placed in layers. The existence of strong frustration (the degree of frustration f = θ_CW_/T_N_∼33.5), caused by J_2_, J_4,_ and J_6_ exchanges, makes it difficult to predict the resulting magnetic ground state. According to DFT calculations, spin configuration E9 has the lowest energy among all studied configurations (Figure 6), regardless of the chosen exchange–correlation functional and U potential. The spins of the Cr2 sites are arranged so that each chromium chain has zero total magnetic moment. We performed DFT calculations with other configurations, where Cr2 spins yield nonzero total magnetic moments of the given chain but maintain overall zero total magnetization. E9 proved to be the lowest energy configuration. The most frustrated chromium site was Cr2, which was connected by magnetic exchanges with eight neighboring Cr1 sites by J_2_, J_4_ and J_6_ exchanges, all of which were frustrated.

The coexistence of several frustrating exchange interactions could allow for lowering the energy of the collinear magnetic subsystem by the canting of magnetic moments. The noncollinearity of the magnetic subsystem can also be promoted by magnetocrystalline anisotropy due to the spin–orbit interaction and the antisymmetric Dzyaloshinskii–Moriya interactions. In an ideal octahedral environment, the 3*d* orbitals split into a higher energy doublet, e_g_, and a lower energy triplet, t_2g_. Thus, Cr^3+^ ions in an octahedral environment are most often in the high-spin state S = 3/2, where all electrons occupy *t*_2g_ orbitals according to Hund’s rule. The absence of orbital degeneracy in the [CrO_6_] octahedron leads to the freezing of the orbital magnetic moment, which is a characteristic feature of 3*d* elements. However, in the Cr_3_BO_6_ structure, the octahedra are strongly distorted, which can lead to unfreezing of the orbital momentum and spin–orbit interaction. Unfortunately, the complexity of the structure of the Cr_3_BO_6_ magnetic subsystem does not enable a more reasonable calculation of the exchange integrals at this stage.

## 2. Discussion

A peculiar feature of Cr_3_BO_6_ with the norbergite structure is the coexistence of long-range antiferromagnetic order based on Cr2 ions with spin-singlets based on Cr1–Cr1 dimers of *S* = 3/2 spins. A similar case was recently discussed for mixed-valence copper compound Pb_2_Cu_10_O_4_(SeO_3_)_4_Cl_7_, which hosts a magnetic network constituted by Cu–Cu dimers and Cu_7_ cluster of corner-sharing Cu_4_ tetrahedra where each Cu_7_ cluster has a *S* = 3/2 spin arrangement in the ground state [27]. Despite the abundance of magnetically active ions, the latter compound orders antiferromagnetically at rather low temperatures and evidences the sequence of spin-flop transition and 1/3 plateau formation at low temperatures. 

The 1/3 magnetization plateau observed for Pb_2_Cu_10_O_4_(SeO_3_)_4_Cl_7_ is explained by the field-induced flip of every second (Cu^2+^)_7_ cluster within a unit cell. The situation in Cr_3_BO_6_ seems to be much simpler; i.e., 1/3 magnetization plateau corresponds to the percentage of magnetically active Cr2 ions, while two-thirds of chromium ions are hidden within magnetically silent dimers. A very large scale of exchange interactions in Cr_3_BO_6_ (Table 2) prevents observation of such a plateau. At µ_0_*H* = 9 T, the magnetization *M* reaches about one-quarter of the value expected at the plateau region.

Recently, many cases of magnetically silent spin-singlet ground state formation have been reported, mostly in low-dimensional copper compounds. Many more frustrated spin systems arrive at magnetically ordered ground states at low temperatures. Cases of the coexistence of spin-singlets with long-range magnetic order are rare. In this sense, the Cr_3_BO_6_ system is unique among chromium compounds.

## 3. Conclusions

In summary, we carried out DFT studies of magnetic exchange interactions and compared the results with experimental data on magnetic properties to predict the magnetic subsystem of chromium borate Cr_3_BO_6_ with the norbergite structure. Considering only magnetic exchange interactions between the nearest chromium ions, the magnetic subsystem of Cr_3_BO_6_ can be described using six different exchange integrals: J_1_–J_6_. Considering the hierarchy of exchange interactions based only on their amplitudes, the magnetic structure is based on a system of interacting AFM dimers via J_1_, connected in chains by J_3_. The existence of strong frustration caused by J_2_, J_4_, and J_6_ exchanges makes it difficult to predict the resulting magnetic ground state. The coexistence of several frustrating exchange interactions makes it possible to predict that the collinear magnetic subsystem will tend to lower its energy due to the canting of magnetic moments. Also, the noncollinearity of the magnetic subsystem can be promoted by magnetocrystalline anisotropy due to the spin–orbit interaction. A detailed experimental study of the magnetic subsystem of the chromium borate Cr_3_BO_6_ remains an urgent task.

## Figures and Tables

**Figure 1 materials-16-07662-f001:**
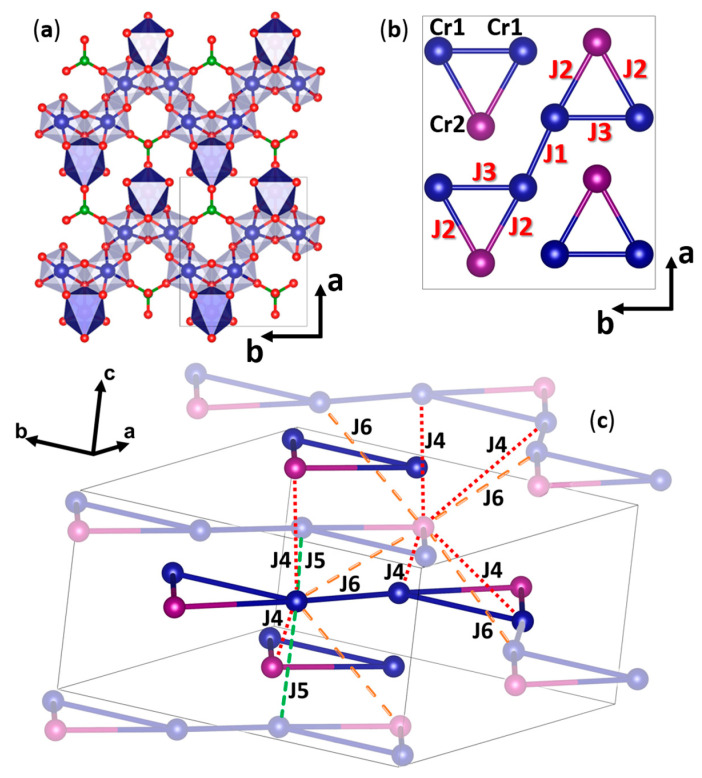
Crystal structure of chromium borate Cr_3_BO_6_ with norbergite structure. (**a**) *ab*-plane of crystal structure. There are two positions of the chromium atom in the structure—8d (Cr1—blue, light octahedra) and 4c (Cr2—purple, dark octahedra), oxygen ions are shown in red, boron ions—in green. (**b**) Schematic representation of the fragment of the magnetic subsystem of Cr_3_BO_6_—triangular zigzag chains of chromium cations. J_1_–J_3_ exchange interactions are shown. (**c**) Mutual arrangement of the zigzag chains and interlayer exchange interactions J_4_–J_6_ in the Cr_3_BO_6_ structure. Atoms outside the unit cell are lightened. Image has been produced using Vesta software [16].

**Figure 2 materials-16-07662-f002:**
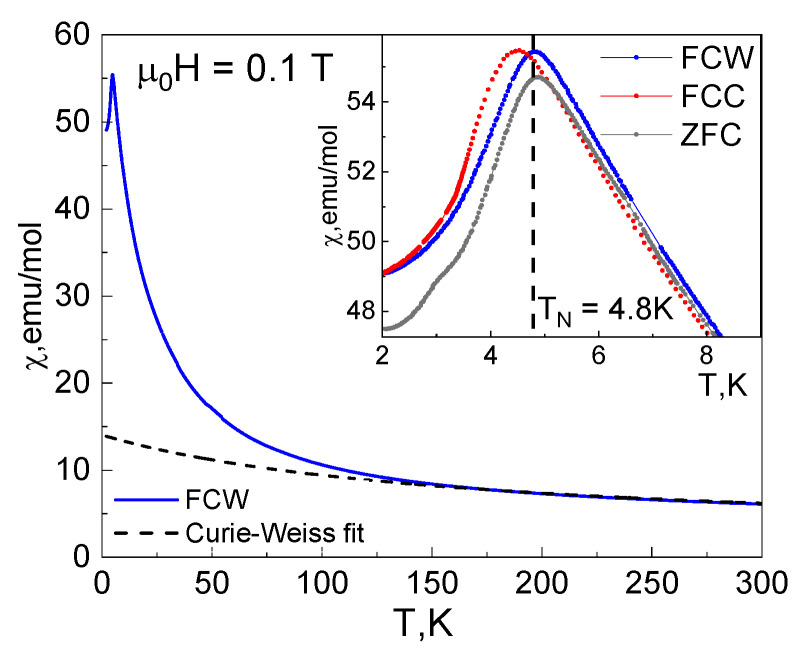
The temperature dependencies of reduced magnetization, χ(*T*) = *M*/*H* in Cr_3_BO_6_, measured at µ_0_*H* = 0.1 T. The inset represents χ(*T*) curves measured according to zero-field-cooled (ZFC), field-cooled-warming (FCW), and field-cooled-cooling (FCC) protocols.

**Figure 3 materials-16-07662-f003:**
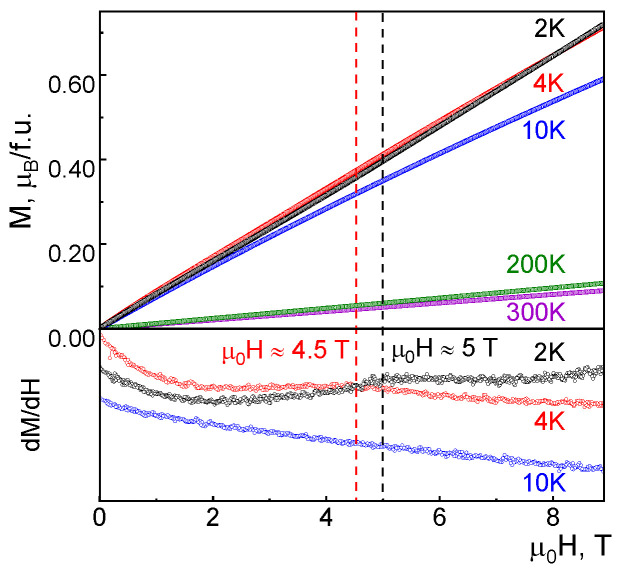
The field dependencies of magnetization of Cr_3_BO_6_ at 2, 4, 10, 200 and 300 K. Lower panel shows the derivatives, *dM*/*dH*, of the low temperature *M*(*H*) curves.

**Figure 4 materials-16-07662-f004:**
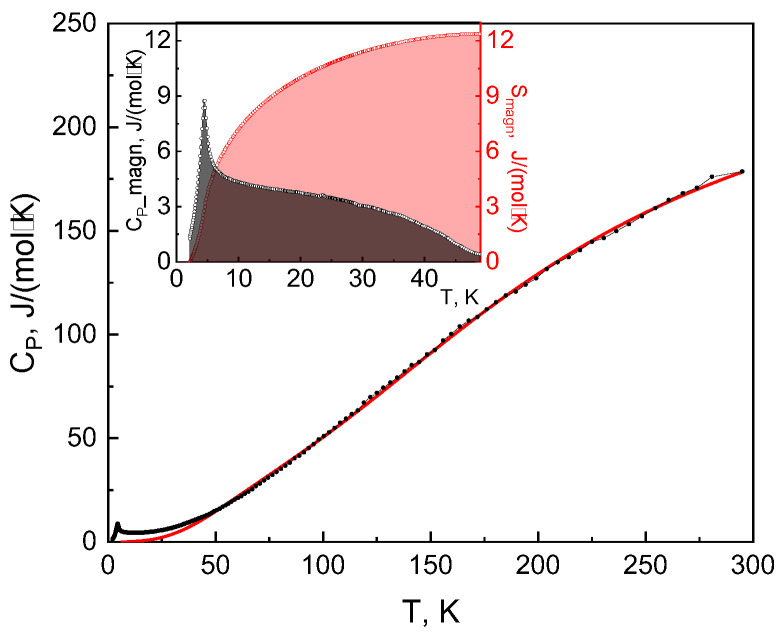
Temperature dependence of specific heat in Cr_3_BO_6_. Solid red line represents a phonon contribution. Inset: temperature dependencies of magnetic specific heat C_magn_ and the magnetic entropy S_magn_.

**Figure 5 materials-16-07662-f005:**
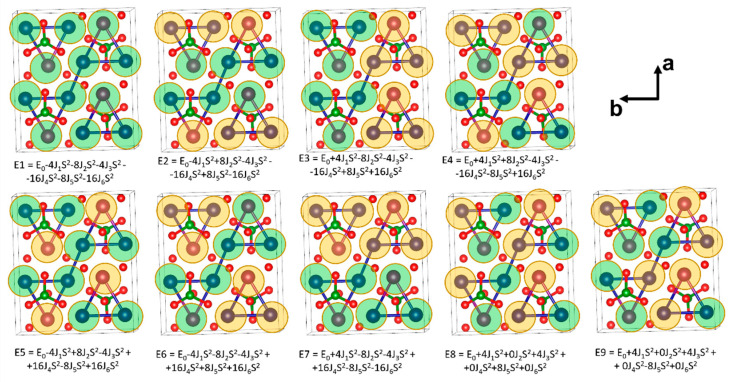
Magnetic sublattice configurations of chromium borate Cr_3_BO_6_ used for theoretical calculations. Green circles indicate spin-up configuration on the ion site, yellow—spin down.

**Figure 6 materials-16-07662-f006:**
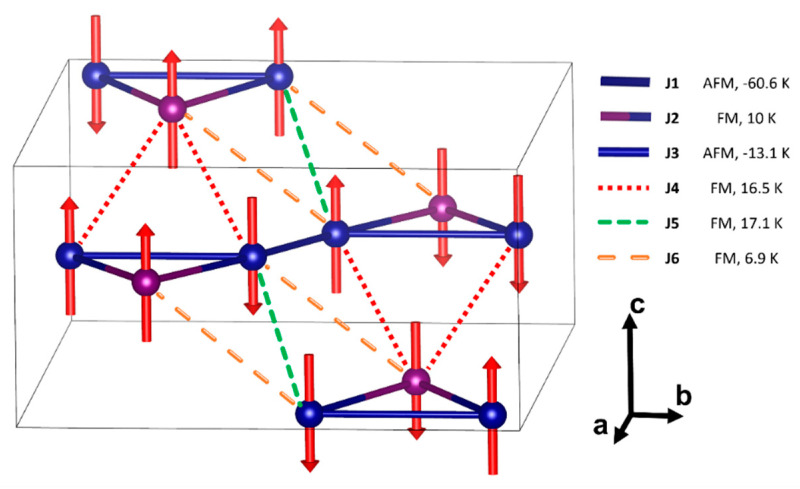
Scheme of magnetic exchange interactions in Cr_3_BO_6_ and proposed collinear magnetic model. Solid lines are related to magnetic interactions in one plane; dashed lines—out of plane.

**Table 1 materials-16-07662-t001:** Inequivalent positions of Cr atoms and their exchange paths.

Atom	Cr1		Cr2	
Exchange	Neighbor (s)	Distance (s), Å	Neighbor (s)	Distance (s), Å
J_1_	Cr1 (in plane)	2.78	—	—
J_2_	Cr2 (in plane)	3.13	2 × Cr1 (in plane)	3.13
J_3_	Cr1 (in plane)	3.05	—	—
J_4_	2 × Cr2	3.40; 3.56	4 × Cr1	3.40; 3.56
J_5_	2 × Cr1	3.47	—	—
J_6_	2 × Cr2	3.45; 3.52	4 × Cr1	3.45; 3.52

**Table 2 materials-16-07662-t002:** Calculated magnetic exchange interactions depending on different calculation parameters.

Exchange, K	PBE, U = 4 eV	PBEsol, U = 4 eV	PBE, U = 6.5 eV	PBEsol, U = 6.5 eV
J_1_	−111.4	−111	−64.4	−60.7
J_2_	−15.1	−7.7	2.5	10.0
J_3_	−7.3	−8.9	−12.1	−13.1
J_4_	8.3	12.2	13.2	16.5
J_5_	−2.15	4.8	10.5	17.1
J_6_	8.5	9.8	6.3	6.9

## Data Availability

Data are contained within the article.

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
