# Peer review of "Thermodynamic Properties and DFT Study on Highly Frustrated Cr_3_BO_6_: Coexistence of Spin-Singlets with Long-Range Magnetic Order"

_materials, 2023, doi:10.3390/ma16247662_

Round 1

Reviewer 1 Report

Comments and Suggestions for Authors This is the first detailed experimental study of the magnetic properties of Cr3BO6. The results suggest that Cr3BO6 has a complex magnetic structure and its magnetic properties arise from strong magnetic frustration. The measurement of specific heat was also performed for the first time for this compound. For this reason, the work is novel and original. In addition, the Authors support their experimental results with density functional calculations. Moreover, the main conclusion present in the title on the coexistence of spin-singlets with long-range antiferromagnetic ordering caused by strong magnetic frustration is very interesting. In general, frustrated magnets are intensively investigated nowadays due to many interesting phenomena e.g. magnetic skyrmions, unconventional anomalous or topological Hall effect and much more. For this reason, the present work could be interesting for a wide range of scientists. The paper is well-written and contains new and interesting results. So I recommend the manuscript for publication after some minor corrections: 1) Some references were omitted:  Figure 1 was generated using the VESTA package and proper references should be given as mentioned in the VESTA License.  In DFT section lacks references to the Quantum Espresso package - for details see "Terms of use" in the latest QE User's guide.  References to PBE and PBEsol functionals  PBE - J. P. Perdew, K. Burke, M. Ernzerhof, Phys. Rev. Lett. 77 (1996) 3865  PBEsol - J. P. Perdew, A. Ruzsinszky, G. I. Csonka, O. A. Vydrov, G. E. Scuseria, L. A. Constantin, X. Zhou, K. Burke, Phys. Rev. Lett. 100 (2008) 136406 2) Was the value of the Hubbard U parameter obtained from the DFPT calculation determined self-consistently? If not, the Authors should try to use a self-consistent approach.

Reviewer 2 Report

Comments and Suggestions for Authors

The authors report a combination study of experimental and theoretical on the peculiar magnetic behaviour of Cr3BO6. Particularly novel in their work is the explicit demonstration from magnetic measurements that Cr3BO6 has a high f-value, and the proof from theoretical calculations that of the two types of Cr sites (Cr1 and Cr2), the one (Cr1) that contributes strongly to frustration. These points are worthy of publication. However, the overall presentation is poor and needs revision.

1.      Their English has to be proofread by a native speaker. There are so many grammatical errors, including in the title.

2.      Although the title only mentions DFT calculations, the paper is a collaboration study of experiment and theory. It is a more interesting paper if it is mentioned that there is a collaboration.

3.      The explanation of the calculation model is difficult to understand. Fig. 1(c) illustrates the calculated unit cell and the magnetic coupling within it, but the structure cannot be found in the crystal structure, Fig. 1(a). It is necessary to explain in the diagram which part of Fig. 1(a) needs to be cut out to obtain the model in Fig. 1(c). They state that the crystal structure (X-ray structure) obtained experimentally was taken directly as the computational model, so there should be a structure in Fig. 1(a) corresponding to Fig. 1(c) (are we looking at different orientations?). In addition, the specific coordinates the calculated model needs to be provided in a universal format (e.g. cif file).

4.      The calculation conditions for the DFT were stated in the Results section. The first and second paragraphs of the DFT calculation section should be stated in the Method section.

5.      To determine the values of E0,J1-J6 they calculated eight spin configurations. Why not seven: I think are seven equations (spin configurations) sufficient for seven variables (E0,J1-J6)?

6.      Their calculated spin configuration (Fig. 5) does not include the ground spin configuration of Cr1 (Fig. 6) that they claim; the results of a spin configuration satisfying Fig. 6 should also be shown.

Comments on the Quality of English Language

There are so many grammatical errors, including in the title.

Reviewer 3 Report

Comments and Suggestions for Authors

The referee report is in the attached pdf file. 

Round 2

Reviewer 2 Report

Comments and Suggestions for Authors

The manuscript has been revised carefully by the authors. The paper is well presented, and it will be published without any additional reviews.

Comment: The sentence "The experimental crystal structure data on chromium borate without additional relaxation was used in the calculations." needs references. Refs. [10] and [12] are suitable.